# Deep learning for automatic tumour segmentation in PET/CT images of patients with head and neck cancers

**Yngve Mardal Moe**[1]                                      YNGVE.MARDAL.MOE@NMBU.NO
**Aurora Rosvoll Groendahl**[1]
**Martine Mulstad**[1]
**Oliver Tomic**[1]
**Ulf Indahl**[1]
**Einar Dale**[2]
**Eirik Malinen**[3,4]
**Cecilia Marie Futsaether**[1]

[1] *Faculty of Science and Technology, Norwegian University of Life Sciences, Aas, Norway.*

[2] *Department of Oncology, Oslo University Hospital, Oslo, Norway.*

[3] *Department of Medical Physics, Oslo University Hospital, Oslo, Norway.*

[4] *Department of Physics, University of Oslo, Oslo, Norway.*

## Abstract

An automatic segmentation algorithm for delineation of the gross tumour volume and pathologic lymph nodes of head and neck cancers in PET/CT images is described. The proposed algorithm is based on a convolutional neural network using the U-Net architecture. Several model hyperparameters were explored and the model performance in terms of the Dice similarity coefficient was validated on images from 15 patients. A separate test set consisting of images from 40 patients was used to assess the generalisability of the algorithm. The performance on the test set showed close-to-oncologist level delineations as measured by the Dice coefficient (CT: $0.65 \pm 0.17$, PET: $0.71 \pm 0.12$, PET/CT: $0.75 \pm 0.12$).

## 1. Introduction

Radiotherapy (RT) is standard treatment for patients with inoperable head and neck cancers (HNC). An essential part of RT planning is delineation of tumours, metastatic lymph nodes and organs at risk. This task is done manually, which is time-consuming, labour-intensive and prone to inter- and intra-observer differences (Gudi et al., 2017). Finding robust and precise methods to automate delineation is, therefore, of great importance.

In this work, we used a deep convolutional neural network to automatically delineate the gross tumour volume (GTV) and pathologic lymph nodes in patients with HNC.

## 2. Methods

Oncologist delineations of gross tumour volume and pathologic lymph nodes were extracted from 197 HNC patients planned for radiotherapy at Oslo University Hospital between January 2007 and December 2013. FDG-PET co-registered to contrast enhanced CT images were available for all patients. The dataset was split into a training (142 patients), validation (15 patients) and test (40 patients) set, stratified by tumour T-stage. The Regional Ethics Committee approved the study.

The ground truth segmentation masks were defined as the union of the oncologist delineated GTV and pathologic lymph nodes. Images were cropped to reduce their size whilst retaining a sufficiently large region of interest (ROI) of the head and neck region (see Figure 1). The network was trained on transversal image slices.

A U-Net architecture (Ronneberger et al., 2015) with zero padded convolutions was trained using the Adam optimiser (Kingma and Ba, 2014) with a learning rate of $10^{-4}$ and standard $\beta$ parameters. Batch normalisation (Ioffe and Szegedy, 2015) was used after each activation function to ensure stable training. Cross entropy loss and the Dice loss were tested (Milletari et al., 2016).

Models were trained using solely CT images (CT-only), solely PET images (PET-only) or both CT images and PET images (PET/CT). Benefits of using CT windowing to highlight soft tissue were examined. Window widths of 100 HU and 200 HU and centres of 60 HU and 70 HU were tested.

Models with all hyperparameter combinations were trained twice with different random seeds. The Dice similarity coefficient on the validation set was used for model comparisons. Models with the highest validation Dice coefficient for each image modality were assessed on the test set.

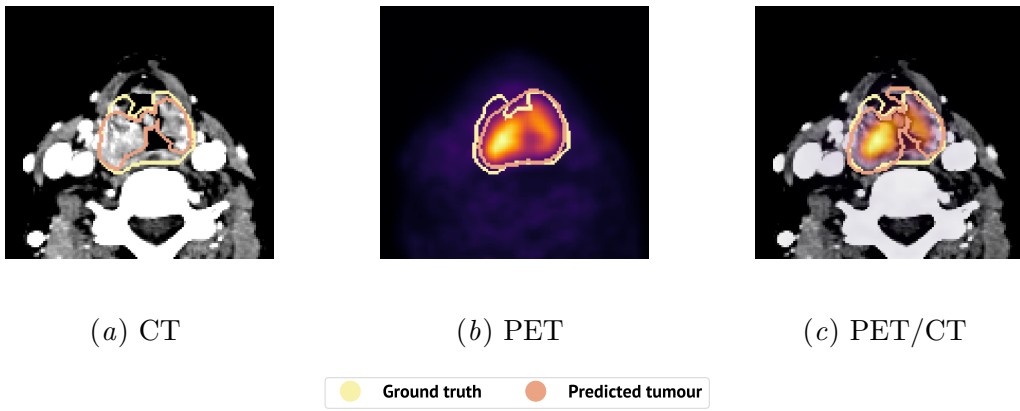

(a) CT             (b) PET             (c) PET/CT

Ground truth    Predicted tumour

Figure 1: The ground truth and predicted segmentation masks for each model.

## 3. Results

Models trained with CT windowing consistently outperformed those trained without. Windowing parameters did not affect performance. Models using both PET and windowed CT images achieved the overall highest performance, followed by PET-only models which outperformed CT-only models. Loss function choice did not affect performance.

Performance metrics obtained on the test set are shown in Table 1. In comparison, a study of interobserver variability in HNC GTV delineation reported a mean Dice coefficient between radiation oncologists of 0.56 and 0.69 for CT-only and PET/CT images, respectively (Gudi et al., 2017).

Figure 1 shows the oncologist ground truth delineations and the CNN predicted segmentations superimposed on input images, for a case with moderate CNN performance (Dice coefficient in the range 0.65-0.72).

Table 1: Performance metrics (mean $\pm$ standard deviation) on the test set.

| **Modality** | Sensitivity | Specificity | Dice | PPV |
|---|---|---|---|---|
| CT-only | $0.68 \pm 0.18$ | $0.99 \pm 0.014$ | $0.65 \pm 0.17$ | $0.67 \pm 0.20$ |
| PET-only | $0.68 \pm 0.17$ | $0.99 \pm 0.003$ | $0.71 \pm 0.12$ | $0.78 \pm 0.11$ |
| PET/CT | $0.74 \pm 0.16$ | $0.99 \pm 0.007$ | $0.75 \pm 0.12$ | $0.78 \pm 0.15$ |

## 4. Conclusion

A deep convolutional neural network using the U-Net architecture was developed for automatic delineation of head and neck cancers in PET and CT images. The CNN required no user involvement or feedback during training and provided close-to-oncologist level performance.

## Acknowledgments

This study was funded by the Norwegian Cancer Society (Grant Number 160907-2014 and 182672-2016). We thank Disruptive Engineering and Eik Ideverksted for letting us use their hardware.

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
