# OpenReview forum: "Deep learning for automatic tumour segmentation in PET/CT images of patients with head and neck cancers"
_MIDL.io/2019/Conference/Abstract — MIDL Abstract 2019_

### Official Review · AnonReviewer2 · 2019-04-29
**Strong application paper with very thorough methodology**

**Rating:** 4
**Confidence:** 3

**Review:**

This application paper tackles the problem of automatic head and neck cancer tumor segmentation in PET and CT scans. The authors use a standard U-Net as neural network, and evaluate the effect of different parameters:
- Choice of modality
- Windowing or no windowing for the CT modality
- Choice of loss function
- Various hyper-parameters

The authors split their original dataset of 197 patients into training set (142), validation (15 patients) and test (40 patients). They report results on the test set (with also standard deviation), and perform several runs with different network initialization. The conclusion is that PET modality alone is doing better than CT alone, but that PET+CT is still slightly better. Also, CT windowing definitely improve performances.

For comparison, inter-observer variability (same data annotated by another oncologist) is reported, showing that the proposed method reach results similar to human overlap.

Minor:
- While pre-processing the data, is the cropping centered around the object ? Shifting randomly the cropped region can avoid some bias wrt the tumor location
- It is not clear if the authors report the best of the two runs (with different seeds), or an average of the two

---

### Official Review · AnonReviewer1 · 2019-05-01
**Good validation study of tumour segmentation in PET/CT images of head and neck cancers using a CNN**

**Rating:** 3
**Confidence:** 2

**Review:**

* The authors use the u-net architecture and experiment with hyperparameters such as training loss functions, CT images with and without windowing, different window widths and centers.
* Training with both CT and PET is compared to training with either of them alone.
* Experiments are repeated twice and mean and standard deviations are reported.
* Results indicate that the CNN performance is on par with inter-observer variability in expert human segmentations.
* It would have been nicer if some more quantitative results had been presented - for instance, the authors say that training with CT windowing helps, but by how much?

Overall, although there is no methodological novelty in the abstract, it offers a good validation study for the dataset in question.

---

### Decision · Program_Chairs · 2019-05-06
**Acceptance Decision**

Accept